# Meta Learning with Relational Information for Short Sequences

**Yujia Xie**
College of Computing, Georgia Tech
Xie.Yujia000@gmail.com

**Haoming Jiang**
College of Engineering, Georgia Tech
jianghm@gatech.edu

**Feng Liu**
Florida Atlantic University
FLIU2016@fau.edu

**Tuo Zhao**
College of Engineering, Georgia Tech
tuo.zhao@isye.gatech.edu

**Hongyuan Zha**[0]
Institute for Data and Decision Analytics, the Chinese University of Hong Kong, Shenzhen
Shenzhen Institute of Artificial Intelligence and Robotics for Society
zhahy@cuhk.edu.cn

## Abstract

This paper proposes a new meta-learning method – named HARMLESS (HAwkes Relational Meta LEarning method for Short Sequences) for learning heterogeneous point process models from short event sequence data along with a relational network. Specifically, we propose a hierarchical Bayesian mixture Hawkes process model, which naturally incorporates the relational information among sequences into point process modeling. Compared with existing methods, our model can capture the underlying mixed-community patterns of the relational network, which simultaneously encourages knowledge sharing among sequences and facilitates adaptive learning for each individual sequence. We further propose an efficient stochastic variational meta expectation maximization algorithm that can scale to large problems. Numerical experiments on both synthetic and real data show that HARMLESS outperforms existing methods in terms of predicting the future events.

## 1 Introduction

Event sequence data naturally arises in analyzing the temporal behavior of real world subjects (Cleeremans and McClelland, 1991). These sequences often contain rich information, which can predict the future evolution of the subjects. For example, the timestamps of tweets of a twitter user reflect his activeness and certain state of mind, and can be used to show when he will tweet next time (Kobayashi and Lambiotte, 2016). The job hopping history of a person usually suggests when he will hop next time (Xu et al., 2017b). Unlike usual sequential data such as text data, event sequences are always asynchronous and tend to be noisy (Ross et al., 1996). Therefore specialized algorithms are needed to learn from such data.

In this paper, we are interested in *short* sequences, a type of sequence data that commonly appears in many real-world applications. Such data is usually short for two possible reasons. One is that the event sequences are short in nature, such as the job hopping history. Another is the observation window is narrow. For example, we are interested in the criminal incidents of an area after a specific regulation is published. Moreover, this kind of data usually appears as a collection of sequences, such as the timestamps of many user's tweets. Our goal is to extract information that can predict the occurrence of future events from a large collection of such short sequences.

Many existing literature considers medium-length or long sequences. They first model a sequence as a parametric point process, e.g., Poisson process, Hawkes process or their neural variants, and apply maximum likelihood estimation to find the optimal parameters (Ogata, 1999; Rasmussen, 2013). However, for short sequences, their lengths are insufficient for reliable inference. One remedy is that we treat the collection of short sequences as independent identically distributed realizations of the same point process, since many subjects, e.g., Twitter users, often share similar behaviors. This makes the inference manageable. However, the learned pattern can be highly biased against certain individuals, especially the non-mainstream users, since this method ignores the heterogeneity within the collection.

An alternative is to recast the problem as a multitask learning problem (Zhang and Yang, 2017) – we target at multi-sequence analysis for multi-subjects. For each sequence, we consider a point process model that slightly deviates from a common point process model, i.e., $\widetilde{f}_j = f_0 + f_j$, where $f_0$ is the common model that captures the main effect, $\widetilde{f}_j$ is the model for the $j$-th sequence, and $f_j$ is the relatively small deviation. Such an assumption that there exists a universal common model cross all subjects, however, is still strong, since the subjects' patterns can differ dramatically. For example, the job hopping history of a software engineer and a human resource manager should have distinct characteristics. Furthermore, such method ignores the relationship of the subjects that usually can be revealed by side information. For example, a social network often shows community pattern (Girvan and Newman, 2002) – across the communities the variation of the subjects is large, while within the communities the variation is small. The connections in the social network, such as "follow" or retweet relationship in Twitter data, can provide us valuable information to identify such community pattern, but the aforementioned methods do not take into account such understanding to help analyzing subjects' behavior.

To this end, we propose a HAwkes Relational Meta LEarning method for Short Sequence (HARM-LESS), which can adaptively learn from a collection of short sequence. More specifically, in a social network, each user often has multiple identities (Airoldi et al, 2008). For example, a Twitter user can be both a military fan and a tech fan. Both his tweet history and social connections are based on his identities. Motivated by above facts, we model each sequence as a hierarchical Bayesian mixture of Hawkes processes – the weights of each Hawkes process are determined jointly by the hidden pattern of sequences and the relational information, e.g., social graphs.

We then propose a variational meta expectation maximization algorithm to efficiently perform inference. Different from existing fully bayesian inference methods (Box and Tiao, 2011; Rasmussen, 2013; Xu and Zha, 2017), we make no assumption on the prior distribution of the parameters of Hawkes process. Instead, when inferring for the Hawkes process parameters of the same identity for all the subjects, we perform a model-agnostic adaptation from a common model for this identity (Finn et al. (2017), see section 3 for more details). This is more flexible since it does not restrict to a specific form. We apply HARMLESS to both synthetic and real short event sequences, and achieve competitive performance.

**Notations**: Throughout the paper, the unbold letters denote vectors or scalars, while the bold letters denote the corresponding matrices or sequences. We refer the $k$-th entry of vector $a_i$ as $a_{i,k}$. We refer the $i$-th subject as subject $i$.

## 2   Preliminaries

We briefly introduce Hawkes Process and Model-Agnostic Meta Learning.

**Hawkes processes** (Hawkes, 1971) is a doubly stochastic temporal point process $\mathcal{H}(\theta)$ with conditional intensity function $\lambda = \lambda(t; \theta, \boldsymbol{\tau})$ defined as

$$\lambda(t; \theta, \boldsymbol{\tau}) = \mu + \sum_{\tau^{(j)} < t} g(t - \tau^{(j)}; \xi),$$

where $\theta = \{\mu, \xi\}$, $g$ is the nonnegative impact function with parameter $\xi$, $\mu$ is the base intensity, and $\boldsymbol{\tau} = \{\tau^{(1)}, \tau^{(2)}, \cdots, \tau^{(M)}\}$ are the timestamps of the events occurring in a time interval $[0, t_{\text{end}}]$. Function $g$ indicates how past events affect current intensity. Existing works usually use pre-specified impact functions in parametric form, e.g., the exponential function in Rasmussen (2013); Zhou et al. (2013) and the power-law function in Zhao et al. (2015).

Hawkes process captures an important property of real-world events – self-exciting, i.e., the past events always increase the chance of arrivals of new events. For example, selling a significant quantity

of a stock can precipitate a trading flurry. As a result, Hawkes process has been widely used in many areas, e.g., behavior analysis (Yang and Zha, 2013; Luo et al., 2015), financial analysis (Bacry et al., 2012), and social network analysis (Blundell et al., 2012; Zhou et al., 2013).

**Model-Agnostic Meta Learning** (MAML, Finn et al., 2017) considers a set of tasks $\Gamma = \{\mathcal{T}_1, \mathcal{T}_2, \cdots, \mathcal{T}_N\}$, where each of the tasks only contains a very small amount of data which is not enough to train a model. We want to exploit the shared structure of the tasks, to obtain models that can perform well on each of the tasks. Specifically, MAML seeks to train a common model for all tasks. From optimization perspective, MAML solves the following problem,

$$\min_\theta \sum_{\mathcal{T}_i \in \Gamma} \mathcal{F}_{\mathcal{T}_i}(\widetilde{\theta}_i) \triangleq \min_\theta \sum_{\mathcal{T}_i \in \Gamma} \mathcal{F}_{\mathcal{T}_i}(\theta - \eta \mathcal{D}(\mathcal{F}_{\mathcal{T}_i}, \theta)), \tag{1}$$

where $\mathcal{D}(\cdot, \cdot)$ is an operator, $\mathcal{F}_{\mathcal{T}_i}$ is the loss function of task $\mathcal{T}_i$, $\theta$ is the parameter of the common model, and $\eta$ is the step size. Here, $\mathcal{D}(\mathcal{F}_{\mathcal{T}_i}, \theta)$ represents one or a small number of gradient update of $\theta$. For example, in cases of one gradient step, we take $\mathcal{D}(\mathcal{F}_{\mathcal{T}_i}, \theta) = \nabla_\theta \mathcal{F}_{\mathcal{T}_i}(\theta)$. This optimization problem aims to find the common model that is expected to produce maximally effective behavior on that task after performing update $\theta - \eta \mathcal{D}(\mathcal{F}_{\mathcal{T}_i}, \theta)$.

Solving (1) using gradient descent involves computing the Hessian matrices, which is computationally prohibitive. To alleviate the computational burden, First Order MAML (FOMAML) (Finn et al., 2017) and Reptile (Nichol et al., 2018) are then proposed. FOMAML drops the second order term in the gradient of (1). Reptile further simplifies the computation by relaxing the original update with Hessian as a multi-step stochastic gradient descent updates. All three algorithms can be written in the form of (1) with operator $\mathcal{D}$ defined differently for different methods. Due to space limit, we defer the definition of $\mathcal{D}$ to Appendix B.

# 3  HAwkes Relational Meta LEarning for Short Sequences (HARMLESS)

We next introduce the meta learning method for analyzing short sequences. Suppose we are given a collection of sequences $\boldsymbol{T} = \{\boldsymbol{\tau}_1, \boldsymbol{\tau}_2 \cdots, \boldsymbol{\tau}_N\}$. We also know some extra relational information about the subjects. For example, in social networks, we can have information on who is friend of whom; in criminal data, we have the locations of the crimes, and crimes happen near each other often have Granger causality. Such relational information can be described as a graph $\mathcal{G} = (\mathcal{E}, \mathcal{V})$, where $\mathcal{E}$ is the node set, $\mathcal{V}$ is the edge set. Denote its adjacency matrix as $\boldsymbol{Y}$.

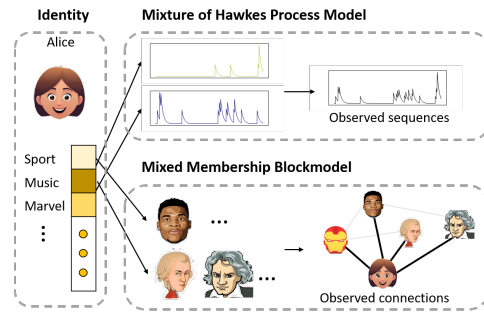

Figure 1: Illustration of the suggested model.

Such social graphs often exhibit community patterns (Girvan and Newman, 2002; Xie et al., 2013). Within the communities the variation of subjects are small, while across the communities the variation is large. Moreover, the communities are overlapping with each other, i.e., each subject may belong to multiple communities and thus have multiple identities. The behaviors of the subject is based on the identities. Motivated by this observation, we first assign each subject a sum-to-one identity proportion vector $\pi_i \in [0, 1]^K$, whose $k$-th entry represents the probability of subject $i$ having the $k$-th identity. In this way, we associate each subject with multiple identities rather than a single identity so that its different aspects is captured, which is more natural and flexible.

For the $k$-th identity of subject $i$, we adopt Hawkes process $\mathcal{H}(\widetilde{\theta}_k^{(i)})$ to model the timestamps of the associated events. Denote the conditional intensity function of $\mathcal{H}(\widetilde{\theta}_k^{(i)})$ as $\lambda(t; \widetilde{\theta}_k^{(i)}, \boldsymbol{\tau}_i)$. For a Hawkes process $\mathcal{H}(\widetilde{\theta}_k^{(i)})$, the likelihood (Laub et al., 2015) of a sequence $\boldsymbol{\tau}_i$ to appear in time interval $[0, t_{\text{end}}]$ is

$$\mathcal{L}(\widetilde{\theta}_k^{(i)}; \boldsymbol{\tau}_i) = \exp\Big( - \int_0^{t_{\text{end}}} \lambda(t; \widetilde{\theta}_k^{(i)}, \boldsymbol{\tau}_i) dt + \sum_{\tau_j < t_{\text{end}}} \log \lambda(\tau_j; \widetilde{\theta}_k^{(i)}, \boldsymbol{\tau}_i) \Big). \tag{2}$$

Here, the parameter $\widetilde{\theta}_k^{(i)}$ is adapted from a common model with parameter $\theta_k$ using a relatively small model-agnostic adaptation, which we will elaborate in next section.

The identity of the $i$-th subject is then a combination of the $K$ identities with identity proportion $\pi_i$, and the models for individual sequences are essentially mixtures of Hawkes process models. Denote $\mathcal{L}_i(\widetilde{\theta}_k^{(i)}) = \mathcal{L}(\widetilde{\theta}_k^{(i)}; \boldsymbol{\tau}_i)$. The likelihood for the $i$-th sequence $\boldsymbol{\tau}_i$ is

$$p(\boldsymbol{\tau}_i) = \sum_{k=1}^{K} \pi_{i,k} \mathcal{L}_i(\widetilde{\theta}_k^{(i)}). \tag{3}$$

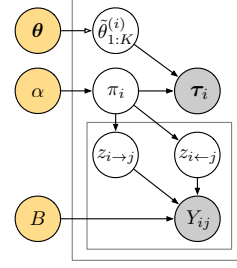

Figure 2: Probabilisitic graph of the suggested model. The yellow nodes are parameters, white nodes are latent variables, and the gray nodes are observed variables. The solid arrows represent probabilistic mapping, while the hollow arrows represent the deterministic mapping.

Moreover, the connections of the subjects are also based on their identities. More specifically, for each connection to happen, one subject $i$ needs to approach another subject $j$, where the identities of subjects $i, j$ are based on $\pi_i, \pi_j$ respectively. Based on this observation, we adopt a Mixed Membership stochastic Blockmodel (MMB) (Airoldi et al., 2008) to model the connections of the subjects. For each subjects pair $(i, j)$, denote the identity of subject $i$ when subject $i$ *approaches* subject $j$ as random variable $z_{i \rightarrow j}$, and the identity of subject $j$ when $j$ *is approached* by $i$ as $z_{i \leftarrow j}$. The probability of $z_{i \rightarrow j}$ represent the $k$-th identity is $\pi_{i,k}$, and the probability of $z_{i \leftarrow j}$ represent the $k$-th identity is $\pi_{j,k}$. The probability of whether subject $i$ and $j$ have a connection is then a function dependent on this two identities - the random variable representing the existence of connection $Y_{ij}$ follows Bernoulli distribution with parameter $z_{i \rightarrow j}^T \boldsymbol{B} z_{i \leftarrow j}$, where $\boldsymbol{B}$ is a learnable parameter.

**Generative process**: The above model can be summarized as the following generative process.

- For each node $i$,
  - Draw a $K$ dimensional identity proportion vector $\pi_i \sim \text{Dirichlet}(\alpha)$.
  - Sample the $i$-th sequence $\boldsymbol{\tau}_i$ from the mixture of Hawkes processes described in (3).
- For each pair of nodes $i$ and $j$,
  - Draw identity indicator for the initiator $z_{i \rightarrow j} \sim \text{Categorical}(\pi_i)$
  - Draw identity indicator for the receiver $z_{i \leftarrow j} \sim \text{Categorical}(\pi_j)$
  - Sample whether there is an edge between $i$ and $j$, $Y_{ij} \sim \text{Bernoulli}(z_{i \rightarrow j}^T \boldsymbol{B} z_{i \leftarrow j})$.

Here, the observed variables are $\boldsymbol{\tau}_i$ and $Y_{ij}$. The parameters are $\alpha$, $\widetilde{\theta}_k^{(i)}$, and $\boldsymbol{B}$. The latent variables are $\pi_i, z_i, z_{i \rightarrow j}$ and $z_{i \leftarrow j}$. The graph model is shown in Figure 2.

## 4 Variational Meta Expectation Maximization

We now introduce our variational meta expectation maximization algorithm. This algorithm incorporates model-agnostic adaptation into variational expectation maximization. In the rest of the paper, we denote $\boldsymbol{z}_{\rightarrow} = \{z_{i \rightarrow j}\}_{i,j=1}^{N}$, $\boldsymbol{z}_{\leftarrow} = \{z_{i \leftarrow j}\}_{i,j=1}^{N}$, $\widetilde{\boldsymbol{\theta}} = \{\widetilde{\theta}_k^{(i)}\}_{i=1,k=1}^{N,K}$.

To ease the computation we add one more latent variable $\boldsymbol{z}$. For the $i$-th sequence, we sample $z_i \sim \text{Categorical}(\pi_i)$. We regard $\boldsymbol{\tau}_i$ as a Hawkes process with parameter $\theta_{z_i}^{(i)}$. Note that this is equivalent to the mixture of Hawkes process described in previous section, since $p(\boldsymbol{\tau}_i) = \sum_k p(z_i = k) \mathcal{L}_i(\widetilde{\theta}_{z_i}^{(i)}) = \sum_k \pi_{i,k} \mathcal{L}_i(\widetilde{\theta}_k^{(i)})$. This can ease the computation because now the update for $\boldsymbol{\pi}$ has close form.

**Variational E step.** The goal is to find an approximation of the following posterior distribution

$$p(\boldsymbol{z}, \boldsymbol{z}_{\rightarrow}, \boldsymbol{z}_{\leftarrow}, \boldsymbol{\pi} | \boldsymbol{T}, \boldsymbol{Y}, \alpha, \widetilde{\boldsymbol{\theta}}, \boldsymbol{B}).$$

We aim to find a distribution $q(\boldsymbol{z}, \boldsymbol{z}_{\rightarrow}, \boldsymbol{z}_{\leftarrow}, \boldsymbol{\pi})$ that minimizes the Kullback-Leibler (KL) divergence to the above posterior distribution. This can be achieved by maximizing the Evidence Lower BOund (ELBO, Blei et al., 2017),

$$\max_{q \in Q} \mathbb{E}_q[\log p(\boldsymbol{z}, \boldsymbol{z}_{\rightarrow}, \boldsymbol{z}_{\leftarrow}, \boldsymbol{\pi}, \boldsymbol{T}, \boldsymbol{Y})] - \mathbb{E}_q[\log q(\boldsymbol{z}, \boldsymbol{z}_{\rightarrow}, \boldsymbol{z}_{\leftarrow}, \boldsymbol{\pi})], \tag{4}$$

where $Q$ is a properly chosen distribution space. We adopt $Q$ as the mean-field variational family, i.e.,

$$q(\boldsymbol{z}, \boldsymbol{z}_\rightarrow, \boldsymbol{z}_\leftarrow, \boldsymbol{\pi}) = q_1(\boldsymbol{\pi}) \prod_i q_2(z_i) \prod_j q_3(z_{i\rightarrow j}) q_4(z_{i\leftarrow j}).$$

where $q_1(\pi_i)$ is the Probability Density Function (PDF) of Dirichlet$(\beta_i)$, $q_2(z_i)$ is the Probability Mass Function (PMF) of Categorical$(\gamma_i)$, $q_3(z_{i\rightarrow j})$ is the PMF of Categorical$(\phi_{ij})$, $q_4(z_{i\leftarrow j})$ is the PMF of Categorical$(\psi_{ij})$, and $\beta_i$, $\gamma_i$, $\phi_{ij}$, $\psi_{ij}$ are variational parameters. By some derivation (see Appendix C for detail), the updates for the variational parameters for solving problem (4) are

$$\beta_{i,k} \leftarrow \alpha_k + \gamma_{i,k} + \sum_{j=1}^N \phi_{ij,k} + \sum_{j=1}^N \psi_{ij,k}, \tag{5}$$

$$\gamma_{i,k} \leftarrow e^{\mathbb{E}_q[\log \pi_{i,k}]} \mathcal{L}_i(\widetilde{\theta}_k^{(i)}), \quad \gamma_{i,k} \leftarrow \frac{\gamma_{i,k}}{\sum_\ell \gamma_{i,\ell}}, \tag{6}$$

$$\phi_{ij,k} \leftarrow e^{\mathbb{E}_q[\log \pi_{i,k}]} \prod_{\ell=1}^K \left( B_{k\ell}^{Y_{ij}} (1 - B_{k\ell})^{1-Y_{ij}} \right)^{\psi_{ij,\ell}}, \quad \phi_{ij,k} \leftarrow \frac{\phi_{ij,k}}{\sum_\ell \phi_{ij,\ell}}, \tag{7}$$

$$\psi_{ij,\ell} \leftarrow e^{\mathbb{E}_q[\log \pi_{j,\ell}]} \prod_{k=1}^K \left( (B_{k\ell})^{Y_{ij}} (1 - B_{k\ell})^{1-Y_{ij}} \right)^{\phi_{ij,k}}, \quad \psi_{ij,\ell} \leftarrow \frac{\psi_{ij,\ell}}{\sum_k \psi_{ij,k}}, \tag{8}$$

where $\mathbb{E}_q[\log \pi_{i,k}] = f_{\mathrm{dg}}(\beta_{i,k}) - f_{\mathrm{dg}}(\sum_\ell \beta_{i,\ell})$, and $f_{\mathrm{dg}}(\cdot)$ is the digamma function.

**Meta inference for $\theta$ and $\widetilde{\theta}$.** Recall that the Hawkes parameter of the $k$-th identity of subject $i$ is $\widetilde{\theta}_k^{(i)}$. Instead of specifying that $\widetilde{\theta}_k^{(i)}$ is sampled from a prior distribution, we adapt the $k$-th common model $\mathcal{H}(\theta_k)$ to sequence $i$ using MAML-type updates,

$$\widetilde{\theta}_k^{(i)} = \theta_k - \eta \mathcal{D}(\log \mathcal{L}_i, \theta_k). \tag{9}$$

Since MAML-type algorithms only perform one or few updates from the common model, the adapted individual models with parameter $\widetilde{\theta}_k^{(i)}$ within one community is close to each other, which meets our expectation that the within-community variation should be small.

The gradient descent step on the log-likelihood of $\boldsymbol{\theta}$ can then be written as

$$\theta_k \leftarrow \theta_k + \eta_{\boldsymbol{\theta}} \nabla_{\theta_k} \left( \sum_{i=1}^N \gamma_{i,k} \log \mathcal{L}_i(\theta_k - \eta \mathcal{D}(\log \mathcal{L}_i, \theta_k)) \right), \tag{10}$$

where $\eta_{\boldsymbol{\theta}}$ is the step size. In this algorithm, we only need to estimate the common models with parameter $\theta_k$, $k = 1, 2, \cdots, K$ instead of all individual models. After we obtain $\theta_k$, the individual models can be easily obtained from Equation (9).

**M step.** We perform maximum likelihood estimation to $\alpha$ and $B$, The updates are as follows,

$$\alpha_k \leftarrow \alpha_k + \eta_\alpha \left( N \big( f_{\mathrm{dg}}(\sum_\ell \alpha_\ell) - f_{\mathrm{dg}}(\alpha_k) \big) + \sum_{i=1}^N \big( f_{\mathrm{dg}}(\beta_{i,k}) - f_{\mathrm{dg}}(\sum_l \beta_{i,\ell}) \big) \right), \tag{11}$$

$$B_{k\ell} \leftarrow \frac{\sum_{ij} Y_{ij} \phi_{ij,k} \psi_{ij,\ell}}{\sum_{ij} \phi_{ij,k} \psi_{ij,\ell}}, \tag{12}$$

where $\eta_\alpha$ is the step size. The detailed derivation can be found in Appendix C.

**Algorithm.** We perform updates (5)-(8), (10)-(12) iteratively until convergence. Note that the updates can also be implemented in stochastic fashion – at each iteration, we sample a mini-batch of sequences, and update their associated parameters (Hoffman et al., 2013).

## 5 Experiments

We first briefly introduce oue experiment settings.

**Impact function.** Following Rasmussen (2013); Zhou et al. (2013), we choose exponential impact function $g(t; \{\delta, \omega\}) = \delta \omega e^{-\omega t}$. The conditional intensity function is

$$\lambda(t; \theta, \boldsymbol{\tau}) = \lambda(t; \{\mu, \delta, \omega\}, \boldsymbol{\tau}) = \mu + \sum_{\tau^{(m)} < t} \delta \omega e^{-\omega(t - \tau^{(m)})}, \tag{13}$$

where $\delta$ and $\omega$ are parameters. Note that each Hawkes process model only contains three parameters, $\mu$, $\delta$, and $\omega$. This is because we target at short sequence. To avoid overfitting, each individual models cannot have too many parameters.

**Regularized likelihood function.** Substitute Eq. (13) into Eq. (2), we have

$$\mathcal{L}(\theta; \boldsymbol{\tau}) = \exp\Big( -\mu t_{\text{end}} - \sum_{\tau^{(n)} < t_{\text{end}}} \Big( \delta(1 - e^{-\omega(t_{\text{end}} - \tau^{(n)})}) - \log\big(\mu + \sum_{\tau^{(m)} < \tau^{(n)}} \delta\omega e^{-\omega(\tau^{(n)} - \tau^{(m)})}\big)\Big)\Big).$$

To keep the parameters non-negative, in practice we replace $\log\mathcal{L}_i(\widetilde{\theta}_k^{(i)})$ with a regularized log-likelihood in update (10),

$$\mathcal{Q}_i(\widetilde{\theta}_k^{(i)}) \triangleq \log\mathcal{L}_i(\widetilde{\theta}_k^{(i)}) + \nu\mathcal{R}(\widetilde{\theta}_k^{(i)}) \triangleq \log\mathcal{L}_i(\widetilde{\theta}_k^{(i)}) + \nu\big(\log(\widetilde{\mu}_k^{(i)}) + \log(\widetilde{\alpha}_k^{(i)}) + \log(\widetilde{\omega}_k^{(i)})\big), \quad (14)$$

where $\widetilde{\theta}_k^{(i)} = \{\widetilde{\mu}_k^{(i)}, \widetilde{\alpha}_k^{(i)}, \widetilde{\omega}_k^{(i)}\}$ is the parameter of the $i$-th Hawkes process of the $k$-th identity, $\nu$ is a regularization coefficient.

**Evaluation metric.** We hold out the last timestamp of each sequence, and split the hold-out timestamps into a validation set and a test set. Another option to do validation and test on event sequence data is to hold out the last two timestamps – we first use the former ones to do validation, then train a new model together with the validation timestamps, and finally report the test result based on the later ones. However, this is not suitable here. This is because the sequences we adopt for experiments are usually very short, sometimes even no more than 5 events in one sequence. As a result, the models trained without or with validation timestamps, e.g., using 3 or 4 timestamps, can be significantly different, which makes the validation procedure very unreliable.

We report the Log-Likelihood (LL) of the test set. More specifically, for each sequence $\boldsymbol{\tau}_i = \{\tau_i^{(1)}, \tau_i^{(2)}, \cdots, \tau_i^{(M_i)}\}$ and parameter $\boldsymbol{\theta}$, the likelihood of next arrival $\tau_i^{(M_i+1)}$ is

$$\widetilde{\mathcal{L}}_i = \sum_{k=1}^K \gamma_{i,k}\lambda\big(\tau_i^{(M_i+1)}; \widetilde{\theta}_k^{(i)}, \boldsymbol{\tau}_i\big)\exp\Big( -\int_{\tau_i^{(M_i)}}^{\tau_i^{(M_i+1)}} \lambda(t; \widetilde{\theta}_k^{(i)}, \boldsymbol{\tau}_i)\,dt\Big).$$

The reported score is the averaged $\log\widetilde{\mathcal{L}}_i$ over subjects. More details can be found in Appendix D.

To estimate of the variance of the estimated log-likelihood, we adopt a multi-split procedure for evaluation. First, we train $m$ candidate models with different hyper-parameters. Then we repeat the following procedure for 30 times: 1). Randomly split a validation set and a test set; 2). Pick a model with highest log-likelihood on the validation set from the $m$ candidate models; 3). Compute the log-likelihood on the test set. Accordingly, we obtain 30 estimates of the log-likelihood. We then report the mean and standard error of the 30 estimates.

**Baselines.** We adopt four baselines as follows.

$\diamond$ *MLE-Sep*: We consider each sequence as a realization of an individual Hawkes process. We perform Maximum Likelihood Estimation (MLE) on each sequence separately, and obtain $N$ models for $N$ sequences.

$\diamond$ *MLE-Com*: We consider all sequences as realizations of the same Hawkes process and learn a common model by MLE.

$\diamond$ *DMHP* (Xu and Zha, 2017): We model sequences as a mixture of Hawkes processes with a Dirichlet distribution as the prior distribution of the mixtures.

$\diamond$ *MTL*: We perform multi-task learning as described in Section 1. More specifically, we adopt Hawkes process model for $f_0$ and $\widetilde{f}_j$. Denote the parameters of $f_0$ and $\widetilde{f}_i$ as $\rho_0 = [\mu_0, \delta_0, \omega_0]^T$ and $\rho_i = [\mu_i, \delta_i, \omega_i]^T$, respectively. We solve

$$\max_{\rho_0, \rho_i} \sum_{i=1}^N \big(\mathcal{Q}_i(\rho_i) + \nu_{\text{mtl}}\|\rho_i - \rho_0\|_2\big),$$

where $\|\rho_i - \rho_0\|_2$ is the $\ell_2$ norm regularizer of $\rho_i - \rho_0$ to promote the difference between $f_0$ and $f_j$ to be small, $\nu_{\text{mtl}}$ is a tuning parameter, and $\mathcal{Q}_i(\cdot)$ is the function defined in Eq. (14).

We would like to remark that another possible baseline is the hierarchical Bayesian model, i.e., we model $\widetilde{\theta}_{1:K}^{(i)}$ to have prior distribution with parameter $\boldsymbol{\theta}$. However, such hierarchical Bayesian model

does not have a closed-form update in variational EM algorithm. Therefore, Markov chain Monte Carlo should be adopted for inference, which is not scalable. For our large scale real graphs we consider here, the time cost is unrealistic. Therefore we leave out this baseline.

**Parameter Tuning.** The detailed tuning procedure and detailed settings of each experiment can be found in Appendix E.

## 5.1 Synthetic Data

**Data generation.** We generate a dataset of 50 nodes with $K = 6$ communities. For each community, we generate Hawkes meta parameters $\theta_k = \{\mu_k, \delta_k, \omega_k\}$ using the following uniform distributions:

$\mu_k \sim \text{Uniform}(0.15, 10),$

$\delta_k \sim \text{Uniform}(0.15, 0.85),$

$\omega_k \sim \text{Uniform}(1, 10).$

We set $\alpha = \mathbf{1}_K$, i.e., the entries of $\alpha$ is all one. Then for the $i$-th node, the identity proportion $\pi_i$ is sampled from Dirichlet$(\alpha)$ and the membership indicator $z_i$ from the corresponding categorical distribution Categorical$(\pi_i)$. Based on $z_i$, we then generate the Hawkes parameters $\widetilde{\theta}_{z_i}^{(i)}$ by adding small perturbation to $\theta_{z_i}$:

$$\widetilde{\mu}_{z_i}^{(i)} \sim \text{N}(\mu_{z_i}, 0.01), \quad \widetilde{\delta}_{z_i}^{(i)} \sim \text{N}(\delta_{z_i}, 0.01), \quad \widetilde{\omega}_{z_i}^{(i)} \sim \text{N}(\omega_{z_i}, 0.05).$$

Table 1: Visualizations of identities by HARMLESS(MAML).

| $S$ | Ground Truth | $K_0 = 3$ | $K_0 = 6$ | $K_0 = 10$ |
|---|---|---|---|---|
| 0.5 | | | | |
| 1.0 | | | | |
| 2.0 | | | | |

The sequence is then sampled based on Hawkes process with parameter $\widetilde{\theta}_{z_i}^{(i)}$ in time interval $[0, 20]$. To ease the tuning we normalize the sequences by dividing by the largest timestamp. We set $B_{k\ell} = \frac{0.5}{N}, \frac{1}{N}, \frac{2}{N}$, for any $k \neq \ell$, and $B_{kk} = \frac{5}{\#\{i \in [1, \cdots, N]: z_i = k\}}$. We sample the graph edges based on $\mathbf{B}$. Denote $S = B_{k\ell} \times N$. The generated graphs are visualized in the second column of Table 1.

**Visualization of communities.** We visualize the communities learned by HARMLESS (MAML) in Table 1. Denote $K_0$ as the number of communities specified in HARMLESS. We adopt $K_0$ colors corresponding to the $K_0$ communities in the graph. The color of each node shown in the Table 1 is the linear combinations of the RGB values of the $K_0$ colors weighted by identity proportions $\pi_i$.

HARMLESS produces reasonable identities even if $K_0$ is mis-specified. If $K_0 < K$, some of the communities would merge. If $K_0 > K$, some of the communities would split.

**Benefit of joint training.** To validate the benefit of joint training on graphs and sequences, we compare HARMLESS result with a two step procedure: We first train an MMB model and obtain the identities, and train HARMLESS (MAML) with fixed identities. In Figure 3 we plot the obtained log-likelihood with respect to $K_0$.

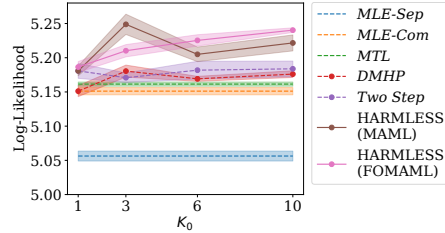

Figure 3: Plot of synthetic data. $S = 1$.

HARMLESS (MAML) consistently achieves larger log-likelihood than the two step procedure. This suggests joint training of graphs and the sequences indeed improve the prediction of future events.

**Log-likelihood with respect to $K_0$.** We also include the results of the baselines and HARMLESS (FOMAML) in Figure 3. The performance of HARMLESS is consistently better than the baselines. Besides, we find the performance HARMLESS (Reptile) is very dependent on the dataset. For this synthetic dataset, Reptile cannot perform well.

## 5.2 Real Data

We adopt four real datasets.

Table 2: Log-likelihood of real datasets.

| Dataset | 911-Calls | LinkedIn | MathOverflow | StackOverflow |
|---|---|---|---|---|
| *MLE-Sep* | $4.0030 \pm 0.3763$ | $0.8419 \pm 0.0251$ | $0.5043 \pm 0.0657$ | $0.2862 \pm 0.0177$ |
| *MLE-Com* | $4.5111 \pm 0.3192$ | $0.8768 \pm 0.0028$ | $1.7805 \pm 0.0345$ | $1.5594 \pm 0.0134$ |
| *DMHP* | $4.4812 \pm 0.3434$ | $0.8348 \pm 0.0030$ | $1.5394 \pm 0.0347$ | $N \backslash A$ |
| *MTL* | $4.4621 \pm 0.3173$ | $0.9270 \pm 0.0027$ | $1.7225 \pm 0.0336$ | $1.4910 \pm 0.0089$ |
| HARMLESS (MAML) | $4.5208 \pm 0.3256$ | $\mathbf{1.4070} \pm 0.0105$ | $1.8563 \pm 0.0345$ | $1.3886 \pm 0.0082$ |
| HARMLESS (FOMAML) | $\mathbf{4.6362} \pm 0.3241$ | $1.0129 \pm 0.004$ | $1.8344 \pm 0.0348$ | $1.5988 \pm 0.0083$ |
| HARMLESS (Reptile) | $4.4929 \pm 0.3503$ | $0.9540 \pm 0.0082$ | $\mathbf{1.8663} \pm 0.0342$ | $\mathbf{1.6017} \pm 0.0097$ |

**911-Calls dataset**: The 911-Calls dataset[1] contains emergency phone call records of fire, traffic and other emergencies for Montgomery County, PA. The county is divided into disjoint areas, each of which has a unique ZIP Code. For each area, the timestamps of emergency phone calls in this area are recorded as an event sequence. We consider each area as a subject, and two subjects are connected if they are adjoint. We finally obtain $57$ subjects and $81$ connections among them. The average length of the sequences is $219.1$.

**LinkedIn dataset**: The LinkedIn dataset (Xu et al., 2017b) contains job hopping records of the users. For each user, her/his check-in timestamps corresponding to different companies are recorded as an event sequence. We consider each user as a subject, and two subjects are connected if the difference in timestamps of two user joined the same company is less than 2 weeks. After removing the singleton subjects, we have $1,369$ subjects and $12,815$ connections among them. The average length of the sequences is $4.9$.

**MathOverflow dataset**: The MathOverflow dataset (Paranjape et al., 2017) contains records of the users posting and answering math questions. We adopt the records from May 2, 2014 to March 6, 2016. For each user, her/his timestamps of answering questions are recorded as an event sequence. We consider each user as a subject, and two subjects are connected if one user answers another user's question. After removing the singleton subjects, we have $1,529$ subjects and $6,937$ connections among them. The average length of the sequences is $11.8$.

**StackOverflow dataset**: StackOverflow is a question and answer site similar to MathOverflow. We adopt the records from November 8, 2015 to December 1, 2015. We construct the sequences and graphs in the same way as MathOverflow. After removing the singleton subjects, we have $13,434$ users and $19,507$ connections among them. The average length of the sequences is $7.7$.

**Result**: The log-likelihood is summarized in Table 2. Note due to Markov chain Monte Carlo is needed for *DMHP*, we cannot get reasonable result for large dataset, i.e., StackOverflow. HARMLESS performs consistently better than the baselines. Since the standard error of the results of 911-Calls dataset are large, we also performed a paired t test. The test shows the difference in log-likelihood between *MLE-Com*, i.e., best of the baselines, and HARMLESS (FOMAML), i.e., best of HARMLESS series, is statistically significant (with $p$ value$= 1.3 \times 10^{-5}$).

### 5.3 Ablation Study

We then perform ablation study using LinkedIn dataset. Three sets of ablation study are considered here:

**Remove inner heterogeneity**: We model each community of sequences using the same parameters, i.e., we set $\widetilde{\theta}_k^{(i)} = \theta_k$.

**Remove grouping**: We set $K = 1$, so that the whole graph is one community. This equivalent to apply the MAML-type algorithms on the sequences directly.

**Remove graph**: We do not consider the graph information, i.e., we remove $z_\rightarrow, z_\leftarrow, Y$ and $B$ from the panel in Figure 2.

Table 3: Results of ablation study.

| Method | Log-Likelihood |
|---|---|
| HARMLESS (MAML) | $\mathbf{1.4070} \pm 0.0105$ |
| HARMLESS (FOMAML) | $1.0129 \pm 0.0042$ |
| HARMLESS (Reptile) | $0.9540 \pm 0.0082$ |
| Remove inner heterogeneity ($K = 3$) | $0.9405 \pm 0.0032$ |
| Remove inner heterogeneity ($K = 5$) | $0.9392 \pm 0.0032$ |
| Remove grouping (MAML) | $0.9432 \pm 0.0031$ |
| Remove grouping (FOMAML) | $0.9376 \pm 0.0031$ |
| Remove grouping (Reptile) | $0.9455 \pm 0.0041$ |
| Remove graph (MAML) | $0.9507 \pm 0.0032$ |
| Remove graph (FOMAML) | $0.9446 \pm 0.0032$ |
| Remove graph (Reptile) | $0.9489 \pm 0.0072$ |

The results in Table 3 suggest that MAML-type adaptation, graph information, and using multiple identities all contribute to the good performance of HARMLESS.

## 6 Discussions

**The setting of meta learning.** The goal of conventional settings of meta learning is to train a model on a set of tasks, so that it can quickly adapt to a new task with only few training samples. Therefore,

people divide the tasks into meta training set and meta test set, where each of the task contains a training set and a test set. The meta model is trained on the meta training set, aiming to minimize the test errors, and validated on the meta test set (Vinyals et al., 2016; Santoro et al., 2016). This setting is designed for supervised learning or reinforcement learning tasks that has accuracy or reward as a clear evaluation metric. Extracting information from the event sequences, however, is essentially an unsupervised learning task. Therefore, we do not separate meta training set and meta test set. Instead, we pull the collection of tasks together, and aim to extract shared information of the collection to help the training of models on individual tasks. Here, each short sequence is a task. We exploit the shared pattern of the collection of the sequences to obtain the models for individual sequences.

**Community Pattern.** The target of Mixed Membership stochastic Blockmodels (MMB) is to identify the communities in a social graph, e.g., the classes in a school. However, real social graphs cannot always be viewed as Erdős-Rényi (ER) graphs assumed by MMB. As argued in Karrer and Newman (2011), for real-world networks, MMB tends to assign nodes with similar degrees to same communities, which is different from the popular interpretation of the community pattern. This property, however, is actually very helpful in our case. As an example, Twitter users that are more active tend to have similar behavior: They tend to make more connections and post tweets more frequently. In contrast, users with very different node degrees often have the tweets histories of different characteristics, and thus should be assigned to different identities. Such property of MMB allows the identities in HARMLESS to represent this non-traditional community patterns in non-ER graphs, i.e., it assigns subjects with various activeness to different communities.

**Mixture of Hawkes processes.** Many existing works adopt mixture of Hawkes process to model sequences that are generated from complicated mechanisms (Yang and Zha, 2013; Li and Zha, 2013; Xu and Zha, 2017). Those works are different from HARMLESS since they do not consider the hierarchical heterogeneity of the sequences, and do not consider the relational information.

**Variants of Hawkes process.** Some attempts have been made to further enhance the flexibility of Hawkes processes. For example, the time-dependent Hawkes process (TiDeH) in Kobayashi and Lambiotte (2016) and the neural network-based Hawkes process (N-SM-MPP) in Mei and Eisner (2017) learn very flexible Hawkes processes with complicated intensity functions. Those models usually have more parameters than vanilla Hawkes processes. For longer sequences, HARMLESS can also be naturally extended to TiDeHs or N-SM-MPP. However, this work focuses on short sequences. These methods are not useful here, since they have too many degrees of freedom.

## Acknowledgement

This work is partially supported by the grant NSF IIS 1717916 and NSF CMMI 1745382. Part of the work done by Hongyuan Zha is supported by Shenzhen Institute of Artificial Intelligence and Robotics for Society, and Shenzhen Research Institute of Big Data.

## Footnotes

[0]Corresponding author. On leave from College of Computing, Georgia Institute of Technology

[1]Data is provided by montcoalert.org.

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
