[Supplementary Material · HARMLESS1248-525-supp.pdf]

## A  Related Works

**Hawkes Process** Hawkes process has long been used to model event sequences (Hawkes, 1971), such as earthquake aftershock sequences (Ogata, 1999), financial transactions (Bauwens and Hautsch, 2009), and events on social networks (Fox et al., 2016; Farajtabar et al., 2017). Its variant, mixture of Hawkes processes model, has also been proved effective in many area (Yang and Zha, 2013; Li and Zha, 2013; Xu and Zha, 2017). In most cases, the learning methodology is variational inference or maximum likelihood estimation (Rasmussen, 2013; Zhou et al., 2013; Zhao et al., 2015). Other possible methods includes least-squares-based method (Eichler et al., 2017), Wiener-Hopf-based methods (Bacry et al., 2012), and cumulants-based methods (Achab et al., 2017).

Instead of predefine an impact function here, some non-parametric methods use discretization or kernel-estimation when learning models (Reynaud-Bouret et al., 2010; Zhou et al., 2013; Hansen et al., 2015). Those methods usually target small datasets, and do not need a good scalability. Recently, some attempts have been made to further enhance the flexibility of Hawkes processes. The time-dependent Hawkes process (TiDeH) in Kobayashi and Lambiotte (2016) and the neural network-based Hawkes process in Mei and Eisner (2017) learn very flexible Hawkes processes with complicated intensity functions. Those methods usually target very long and multi-dimensional sequences, instead of short sequences.

Existing works targeting short sequences is usually in specific cases (Xu et al., 2017a,b), such as the data is censored. However, there is no work targeting general short sequences as we do here.

There are lines of research that involves both point processes and graphs. One is using point process to find the latent graph (Blundell et al., 2012; Linderman and Adams, 2014; Tran et al., 2015). Another one is considering the interaction of the nodes as point process and use it to construct a dynamic graph, instead of the event happens on nodes as we consider here (Farajtabar et al., 2016; Zarezade et al., 2017; Trivedi et al., 2018). These works have vary different aims from our work.

**Meta Learning** Meta learning has been studied since last century (Bengio et al., 1990; Chalmers, 1991). Some works focus on learning the hyperparameters, such as learning rates or initial conditions (Maclaurin et al., 2015). Some works aim to learn a metric so that a simple K nearest neighbors can perform well under such a metric (Koch et al., 2015; Vinyals et al., 2016; Sung et al., 2018; Snell et al., 2017). Some works design specific deep neural networks so that the information of different tasks are memorized and thus the model can easily generalize to new tasks (Santoro et al., 2016; Munkhdalai and Yu, 2017; Ravi and Larochelle, 2016).

Model-Agnostic Meta Learning (MAML) method (Finn et al., 2017) opens another line of research, i.e., it designs an optimization scheme so that the model can fast adapt to new tasks. Reptile (Nichol and Schulman, 2018), a variant of MAML, is proposed to simplify the computation of MAML. None of those works, however, considers the relational information between tasks like our method, which is critical in modeling short sequences.

One interesting line of follow-up works of MAML is connecting MAML with Bayesian inference (Finn et al., 2018; Ravi and Beatson, 2018; Grant et al., 2018). Since HARMLESS combines a Bayesian model with MAML, it has the potential to be rewritten into a pure Bayesian model that has better quantification of uncertainty. We left this for future work.

## B  Definition of Operator $\mathcal{D}$

As we mentioned earlier,

$$\min_{\theta} \sum_{\mathcal{T}_i \in \Gamma} \mathcal{F}_{\mathcal{T}_i}(\widetilde{\theta}_i) = \sum_{\mathcal{T}_i \in \Gamma} \mathcal{F}_{\mathcal{T}_i}(\theta - \eta \mathcal{D}(\mathcal{F}_{\mathcal{T}_i}, \theta))$$

is the loss function for MAML, FOMAML, and Reptile algorithm with different definition of the operator $\mathcal{D}$.

For simplicity, here we define the operator of one gradient step. The cases of few gradient steps can be defined analogously.

For MAML, $\mathcal{D}(\mathcal{F}_{\mathcal{T}_i}, \theta)$ is defined as $\nabla_{\theta}(\mathcal{F}_{\mathcal{T}_i}(\theta))$.

514 For First Order MAML (FOMAML), $\mathcal{D}(\mathcal{F}_{\mathcal{T}_i}, \theta)$ is also defined as $\nabla_\theta(\mathcal{F}_{\mathcal{T}_i}(\theta))$. The difference is
515 that the output of the operator just a value, not a function of $\theta$, i.e., when we solve the gradient of
516 $\mathcal{F}_{\mathcal{T}_i}(\theta - \eta\mathcal{D}(\mathcal{F}_{\mathcal{T}_i}, \theta))$, the gradient does not back-propagate into $\mathcal{D}(\mathcal{F}_{\mathcal{T}_i}, \theta)$.

517 For Reptile, the algorithm of reptile is as follows Nichol and Schulman (2018).

---
**Algorithm 1** Reptile
---
**while** not converged **do**
&emsp;&emsp;Sample task $\mathcal{T}$ with loss $\mathcal{F}_{\mathcal{T}}$;
&emsp;&emsp;$W \leftarrow \text{SGD}(\mathcal{F}_{\mathcal{T}}, \theta, k)$, where $k$ is the number of SGD steps;
&emsp;&emsp;Do the update $\theta \leftarrow \theta - \eta(\theta - W)$;
**end while**
---

518 From the algorithm we can see, operator $\mathcal{D}$ is defined as $\mathcal{D}(\mathcal{F}_{\mathcal{T}}, \theta) = \text{SGD}(\mathcal{F}_{\mathcal{T}}, \theta, 1)$. Similar as
519 FOMAML, computing the gradient also does not back-propagate into $\mathcal{D}(\mathcal{F}_{\mathcal{T}_i}, \theta)$.

## 520   C   Derivation of Variational EM

521 **Preparation** After adding latent variable $z$, the joint distribution is

$$p(\boldsymbol{T}, \boldsymbol{Y}, \boldsymbol{z}, \boldsymbol{z}_\rightarrow, \boldsymbol{z}_\leftarrow, \boldsymbol{\pi}) = p(\boldsymbol{T}|\boldsymbol{z})p(\boldsymbol{Y}|\boldsymbol{z}_\rightarrow, \boldsymbol{z}_\leftarrow)p(\boldsymbol{z}|\boldsymbol{\pi})p(\boldsymbol{z}_\leftarrow|\boldsymbol{\pi})p(\boldsymbol{z}_\rightarrow|\boldsymbol{\pi})p(\boldsymbol{\pi}).$$

522 where

$$p(\boldsymbol{T}|\boldsymbol{z}) = \prod_{i=1}^N \prod_{k=1}^K \left(\mathcal{L}_i(\theta_k - \eta\mathcal{D}(\mathcal{L}_i, \theta_k))\right)^{z_{i,k}},$$

$$p(\boldsymbol{Y}|\boldsymbol{z}_\rightarrow, \boldsymbol{z}_\leftarrow) = \prod_{i=1}^N \prod_{j=1}^N (z_{i\rightarrow j}^T \boldsymbol{B} z_{i\leftarrow j})^{Y_{ij}} (1 - z_{i\rightarrow j}^T \boldsymbol{B} z_{i\leftarrow j})^{1-Y_{ij}}$$

$$p(\boldsymbol{z}|\boldsymbol{\pi}) = \prod_{i=1}^N \prod_{k=1}^K \pi_{i,k}^{z_{i,k}},$$

$$p(\boldsymbol{z}_\rightarrow|\boldsymbol{\pi}) = \prod_{i=1}^N \prod_{j=1}^N \prod_{k=1}^K \pi_{i,k}^{z_{i\rightarrow j,k}},$$

$$p(\boldsymbol{z}_\leftarrow|\boldsymbol{\pi}) = \prod_{i=1}^N \prod_{j=1}^N \prod_{k=1}^K \pi_{j,k}^{z_{i\leftarrow j,k}},$$

$$p(\boldsymbol{\pi}) = \prod_{i=1}^N \text{Dirichlet}(\pi_i|\alpha) = \prod_{i=1}^N C(\alpha) \prod_{k=1}^K \pi_{i,k}^{\alpha-1}.$$

523 Note that in this section we represent $z_i, z_{i\rightarrow j}, z_{i\leftarrow j}$ as one-hot vector, while in the main paper we
524 use scalar $z_i = k$ representing the identities.

525 The posterior distribution is defined as

$$p(\boldsymbol{z}, \boldsymbol{z}_\rightarrow, \boldsymbol{z}_\leftarrow, \boldsymbol{\pi}|\boldsymbol{T}, \boldsymbol{Y}, \alpha, \boldsymbol{\theta}, B).$$

526 We aim to find a distribution $q(\boldsymbol{z}, \boldsymbol{z}_\rightarrow, \boldsymbol{z}_\leftarrow, \boldsymbol{\pi}) \in Q$, such that the Kullback-Leibler (KL) divergence
527 between the above posterior distribution and $q(\boldsymbol{z}, \boldsymbol{z}_\rightarrow, \boldsymbol{z}_\leftarrow, \boldsymbol{\pi})$ is minimized. This can be achieved by
528 maximize the Evidence Lower BOund (ELBO),

$$\mathcal{B}(q) = \mathbb{E}_q[\log p(\boldsymbol{z}, \boldsymbol{z}_\rightarrow, \boldsymbol{z}_\leftarrow, \boldsymbol{\pi}, \boldsymbol{T}, \boldsymbol{Y})] - \mathbb{E}_q[\log q(\boldsymbol{z}, \boldsymbol{z}_\rightarrow, \boldsymbol{z}_\leftarrow, \boldsymbol{\pi})].$$

529 **Variational family** We adopt the mean-field variational family, i.e.,

$$q(\boldsymbol{z}, \boldsymbol{z}_\rightarrow, \boldsymbol{z}_\leftarrow, \boldsymbol{\pi}) = q_1(\boldsymbol{\pi}) \prod_i q_2(z_i) \prod_j q_3(z_{i\rightarrow j})q_4(z_{i\leftarrow j}).$$

530 We pick $q_1(\pi_i)$ as PDF of Dirichlet($\beta$), $q_2(z_i)$ as PDF of Categorical($\gamma_i$), $q_3(z_{i \to j})$ as PDF of
531 Categorical($\phi_{ij}$), $q_4(z_{i \leftarrow j})$ as PDF of Categorical($\psi_{ij}$).

532 **Update for $q_1$** Again, our goal is to maximize

$$\mathcal{B}(q) = \mathbb{E}_q[\log p(\boldsymbol{z}, \boldsymbol{z}_\to, \boldsymbol{z}_\leftarrow, \boldsymbol{\pi}, \boldsymbol{T}, \boldsymbol{Y})] - \mathbb{E}_q[\log q(\boldsymbol{z}, \boldsymbol{z}_\to, \boldsymbol{z}_\leftarrow, \boldsymbol{\pi})].$$

533 Now we focus on $q_1$, and treat $q_2$, $q_3$ and $q_4$ as given. We want to maximize

$$\begin{aligned}
\mathcal{F}_{\boldsymbol{\pi}}(q_1) &= \mathbb{E}_q[\log p(\boldsymbol{z}, \boldsymbol{z}_\to, \boldsymbol{z}_\leftarrow, \boldsymbol{\pi}, \boldsymbol{T}, \boldsymbol{Y})] - \mathbb{E}_q[\log q(\boldsymbol{z}, \boldsymbol{z}_\to, \boldsymbol{z}_\leftarrow, \boldsymbol{\pi})] \\
&= \mathbb{E}_q[\log p(\boldsymbol{T}|\boldsymbol{z}) + \log p(\boldsymbol{Y}|\boldsymbol{z}_\leftarrow, \boldsymbol{z}_\to) + \log p(\boldsymbol{z}|\boldsymbol{\pi}) + \log p(\boldsymbol{z}_\leftarrow|\boldsymbol{\pi}) + \log p(\boldsymbol{z}_\to|\boldsymbol{\pi}) + \log p(\boldsymbol{\pi})] \\
&\quad - \mathbb{E}_{q_1}[\log q_1(\boldsymbol{\pi})] + \text{const} \\
&= \mathbb{E}_q[\log p(\boldsymbol{z}|\boldsymbol{\pi}) + \log p(\boldsymbol{z}_\leftarrow|\boldsymbol{\pi}) + \log p(\boldsymbol{z}_\to|\boldsymbol{\pi}) + \log p(\boldsymbol{\pi})] - \mathbb{E}_{q_1}[\log q_1(\boldsymbol{\pi})] + \text{const} \\
&= \int q_1(\boldsymbol{\pi}) \left( \mathbb{E}_{q_2}[\log p(\boldsymbol{z}|\boldsymbol{\pi}) + \log p(\boldsymbol{z}_\leftarrow|\boldsymbol{\pi}) + \log p(\boldsymbol{z}_\to|\boldsymbol{\pi}) + \log p(\boldsymbol{\pi})] - \log q_1(\boldsymbol{\pi}) \right) d\boldsymbol{\pi} + \text{const}.
\end{aligned}$$

534 Take the derivative,

$$\frac{\delta \mathcal{F}_{\boldsymbol{\pi}}(q_1)}{\delta q_1} = \mathbb{E}_{q_2}[\log p(\boldsymbol{z}|\boldsymbol{\pi}) + \log p(\boldsymbol{z}_\leftarrow|\boldsymbol{\pi}) + \log p(\boldsymbol{z}_\to|\boldsymbol{\pi}) + \log p(\boldsymbol{\pi})] - \log q_1(\boldsymbol{\pi}) - 1 = 0.$$

535 Substitute the expressions of the distributions, after some derivation we get the update for $\beta$ as

$$\beta_{i,k} \leftarrow \alpha_k + \gamma_{i,k} + \sum_{j=1}^N \phi_{ij,k} + \sum_{j=1}^N \psi_{ij,k}. \tag{15}$$

536 **Update for $q_2$** Similarly, we have

$$\begin{aligned}
\mathcal{F}_{\boldsymbol{z}}(q_2) &= \mathbb{E}_q[\log p(\boldsymbol{T}|\boldsymbol{z}) + \log p(\boldsymbol{z}|\boldsymbol{\pi})] - \mathbb{E}_{q_2}[\log q_2(\boldsymbol{z})] + \text{const} \\
&= \int q_2(\boldsymbol{z}) \left( \mathbb{E}_{q_1}[\log p(\boldsymbol{T}|\boldsymbol{\theta}, \boldsymbol{z}) + \log p(\boldsymbol{z}|\boldsymbol{\pi})] - \log q_2(\boldsymbol{z}) \right) d\boldsymbol{z} + \text{const}.
\end{aligned}$$

537 Take the derivative,

$$\frac{\delta \mathcal{F}_{\boldsymbol{z}}(q_2)}{\delta q_2} = \log p(\boldsymbol{T}|\boldsymbol{\theta}, \boldsymbol{z}) + \mathbb{E}_{q_1}[\log p(\boldsymbol{z}|\boldsymbol{\pi})] - \log q_2(\boldsymbol{z}) - 1 = 0.$$

538 After some derivation, we have

$$\gamma_{i,k} \leftarrow \mathcal{L}_i(\theta_k - \eta \mathcal{D}(\mathcal{L}_i, \theta_k)) \exp \left( f_{\text{dg}}(\beta_{i,k}) - f_{\text{dg}}(\sum_\ell \beta_{i,\ell}) \right), \tag{16}$$

$$\gamma_{i,k} \leftarrow \frac{\gamma_{i,k}}{\sum_\ell \gamma_{i,\ell}}, \tag{17}$$

539 where $f_{\text{dg}}$ is the digamma function.

540 **Update for $q_3$ and $q_4$** The derivation of update for $q_3$ and $q_4$ is very similar to the update for $q_2$, so
541 we will not elaborate on that. Readers who are interested might also refer to Airoldi et al. (2008).
542 The updates are

$$\phi_{ij,k} \leftarrow e^{\mathbb{E}_q[\log \pi_{i,k}]} \prod_{\ell=1}^K \left( B_{k\ell}^{Y_{ij}} (1 - B_{k\ell})^{1 - Y_{ij}} \right)^{\psi_{ij,\ell}}, \quad \phi_{ij,k} \leftarrow \frac{\phi_{ij,k}}{\sum_\ell \phi_{ij,\ell}}, \tag{18}$$

$$\psi_{ij,\ell} \leftarrow e^{\mathbb{E}_q[\log \pi_{j,\ell}]} \prod_{k=1}^K \left( (B_{k\ell})^{Y_{ij}} (1 - B_{k\ell})^{1 - Y_{ij}} \right)^{\phi_{ij,k}}, \quad \psi_{ij,k} \leftarrow \frac{\psi_{ij,k}}{\sum_\ell \psi_{ij,\ell}}, \tag{19}$$

543 **Update for $\boldsymbol{\theta}$** We update $\boldsymbol{\theta}$ using gradient ascent. We first pick the terms that is relevant to $\boldsymbol{\theta}$,

$$\begin{aligned}
\mathcal{F}_{\boldsymbol{\theta}}(\boldsymbol{\theta}) &= \mathbb{E}_q[\log p(\boldsymbol{T}|\boldsymbol{\theta}, \boldsymbol{z})] + \text{const} \\
&= \int q_2(\boldsymbol{z})[\log p(\boldsymbol{T}|\boldsymbol{\theta}, \boldsymbol{z})] d\boldsymbol{z} + \text{const} \\
&= \sum_{i=1}^N \sum_{k=1}^K \gamma_{i,k} \log \mathcal{L}_i(\theta_k - \eta \mathcal{D}(\mathcal{L}_i, \theta_k)) + \text{const}.
\end{aligned}$$

So the gradient ascent update is,

$$\boldsymbol{\theta} \leftarrow \boldsymbol{\theta} + \eta_1 \nabla_{\boldsymbol{\theta}} \left( \sum_{i=1}^{N} \sum_{k=1}^{K} \gamma_{i,k} \log \mathcal{L}_i(\theta_k - \eta \mathcal{D}(\mathcal{L}_i, \theta_k)) \right). \tag{20}$$

**Update for $\alpha$ and $B$** From Airoldi et al. (2008), we have the update for $\alpha$ and $\boldsymbol{B}$ as follows

$$\alpha_k \leftarrow \alpha_k + \eta_\alpha \left( N \big( f_{\mathrm{dg}}(\sum_\ell \alpha_\ell) - f_{\mathrm{dg}}(\alpha_k) \big) + \sum_{i=1}^{N} \big( f_{\mathrm{dg}}(\beta_{i,k}) - f_{\mathrm{dg}}(\sum_\ell \beta_{i,\ell}) \big) \right), \tag{21}$$

$$B_{k\ell} \leftarrow \frac{\sum_{ij} Y_{ij} \phi_{ij,k} \psi_{ij,\ell}}{\sum_{ij} \phi_{ij,k} \psi_{ij,\ell}}, \tag{22}$$

# D  Derivation of Evaluation Metric

In this section, we give more details on the evaluate metrics. Specifically, we show how to compute the NLL of the test set. Given a sequence $\boldsymbol{\tau}_i = \{\tau_i^{(1)}, \tau_i^{(2)}, \cdots, \tau_i^{(M_i)}\}$, we would like to predict the timestamp of $\tau_i^{(M_i+1)}$. Here, we use the probability of the arrival at time $\tau_i^{(M_i+1)}$ and no arrival in $[\tau_i^{(M_i)}, \tau_i^{(M_i+1)}]$ given history before $\tau_i^{(M_i)}$ as evaluation metric.

Consider a Hawkes process with parameter $\theta$, the probability density is

$$\mathcal{P}(\theta) = \lambda\big(\tau_i^{(M_i+1)}; \theta, \boldsymbol{\tau}_i)\big) \exp\left( - \int_{\tau_i^{(M_i)}}^{\tau_i^{(M_i+1)}} \lambda(t; \theta, \boldsymbol{\tau}_i)\, dt \right)$$

$$= \big(\mu + \sum_{m=1}^{M_i} \delta\omega e^{-\omega(\tau_i^{(M_i+1)} - \tau_i^{(m)})}\big) \exp\left( -\mu(\tau_i^{(M_i+1)} - \tau_i^{(M_i)}) - \delta(1 - e^{-\omega(\tau_i^{(M_i+1)} - \tau_i^{(M_i)})}) \right).$$

In the generative process, for subject $i$, we first sample $z_i$, then use parameter $\widetilde{\theta}_{z_i}^{(i)} = \theta_{z_i} - \eta \mathcal{D}(\mathcal{L}_i, \theta_{z_i})$. The posterior distribution of $z_i$ is $q_2(z_i)$, i.e., Categorical$(\gamma_i)$. Therefore we have

$$\mathbb{P}(z_i = k) = \gamma_{i,k}.$$

So the likelihood of next arrival $\tau_i^{(M_i+1)}$ is

$$\widetilde{\mathcal{L}}_i = \sum_{k=1}^{K} \mathbb{P}(z_i = k) \mathbb{P}(\text{next arrival is } \tau_i^{(M_i+1)} | \text{ Hawkes model with } \theta_k)$$

$$= \sum_{k=1}^{K} \gamma_{i,k} \mathcal{P}(\widetilde{\theta}_k^{(i)}).$$

And then we sum $\widetilde{\mathcal{L}}_i$ over every subject.

# E  Detailed Settings of the Experiments

Note that we can also adopt a non-informative $\alpha$ instead of updating it in every iteration. After some trial experiments, we find setting $\alpha = \mathbf{1}_K$ is numerically more stable than updating it in every iteration. Therefore we adopt $\alpha = \mathbf{1}_K$ in the following experiments.

Besides, we find that $\nu$ causes nearly no effect to the result when varying from $10^{-10}$ to $10^{-1}$. We fix it as $10^{-2}$.

## E.1  Synthetic Dataset

Both the baselines and our proposed methods are fine tuned. We first perform a coarse grid search to find hyper-parameters for all methods. The grid search finds learning rate from $1 \times 10^{-7}$ to $1$ for both inner and outer updates. To perform the multi-split procedure, all hyper-parameters are then selected in the following range listed in Table 4 and Table 5. For each range, we perform experiment on three values: the lower one, the upper one, and the middle one. Method *MTL* adopt $\nu_{\mathrm{mtl}} = 0.1$.

Table 4: Learning rates of experiments.

| $K_0$ | | 1 | 3 | 6 | 10 |
|---|---|---|---|---|---|
| DMHP | lr. | $1\pm.1\times10^{-3}$ | $3\pm.1\times10^{-3}$ | $6.5\pm.1\times10^{-3}$ | $7\pm.1\times10^{-3}$ |
| Two Step | inner lr. | $1\pm.1\times10^{-5}$ | $5\pm.1\times10^{-5}$ | $5\pm.1\times10^{-5}$ | $1\pm.1\times10^{-4}$ |
| | outer lr. | $1\pm.1\times10^{-3}$ | $1\pm.1\times10^{-2}$ | $1.5\pm.1\times10^{-2}$ | $1\pm.1\times10^{-2}$ |
| HARMLESS | inner lr. | $5\pm.1\times10^{-5}$ | $5\pm.1\times10^{-6}$ | $2\pm.1\times10^{-4}$ | $7\pm.1\times10^{-5}$ |
| (MAML) | outer lr. | $6\pm.1\times10^{-4}$ | $2\pm.1\times10^{-4}$ | $6\pm.1\times10^{-5}$ | $4.5\pm.1\times10^{-6}$ |
| HARMLESS | inner lr. | $5\pm.1\times10^{-4}$ | $1\pm.1\times10^{-5}$ | $3\pm.1\times10^{-5}$ | $1.5\pm.1\times10^{-6}$ |
| (FOMAML) | outer lr. | $6\pm.1\times10^{-4}$ | $2\pm.1\times10^{-4}$ | $6\pm.1\times10^{-5}$ | $4.5\pm.1\times10^{-6}$ |

Table 5: Learning rates of baseline experiments.

| Method | Learning Rate |
|---|---|
| MLE-Sep | $5\pm.1\times\times10^{-5}$ |
| MLE-Com | $1\pm.1\times10^{-3}$ |
| MTL | $1\pm.1\times10^{-3}$ |

## E.2 Real Datasets

In this section, we introduce the experimental detail of the real datasets. We run our experiment with same inner and outer learning rate, denoted by $\eta$. For simplicity, we also set $\eta = \eta_\alpha = \eta_{\boldsymbol{\theta}}$, and search over $\{10^{-4}, 10^{-3}, 10^{-2}, 10^{-1}\} \otimes \{1, 2, 3, 4, 5\}$, where the element-wise product of two sets is defined as $A \otimes B = \{ab | a \in A, b \in B\}$. We search $K \in \{2, 3, 5\}$ and $\nu_{\mathrm{mtl}}$ in range $\{0.1, 0.01, 0.001\}$. We perform grid search over the hyper-parameters, and obtain the candidate models. Then we perform multi-split procedure.

Because StackOverflow dataset is very large, it is too expensive to perform grid search. To accommodate this, we first split a validation set and a test set, then performing hyper-parameter search by flipping. Each experiment of StackOverflow dataset is run under 5 different settings.

In Table 6 we report one of the models that is picked by multi-split procedure. We remark that in most cases, the procedure picks only one model repeatedly.

Table 6: Settings of experiments.

| data type | 911-Calls | Linkedin | MathOverflow | StackOverflow |
|---|---|---|---|---|
| Baseline 1 | $\eta = 4 \times 10^{-4}$ | $\eta = 1 \times 10^{-3}$ | $\eta = 5 \times 10^{-4}$ | $\eta = 5 \times 10^{-4}$ |
| Baseline 2 | $\eta = 3 \times 10^{-}$ | $\eta = 5 \times 10^{-3}$ | $\eta = 1 \times 10^{-3}$ | $\eta = 1 \times 10^{-3}$ |
| MTL | $\eta = 3 \times 10^{-5}, \nu_{\mathrm{mtl}} = 0.1$ | $\eta = 1 \times 10^{-2}, \nu_{\mathrm{mtl}} = 0.1$ | $\eta = 4 \times 10^{-4}, \nu_{\mathrm{mtl}} = 0.1$ | $\eta = 5 \times 10^{-4}, \nu_{\mathrm{mtl}} = 0.1$ |
| DMHP | $\eta = 3 \times 10^{-5}, K = 2$ | $\eta = 1 \times 10^{-3}, K = 3$ | $\eta = 4 \times 10^{-3}, K = 3$ | $N \backslash A$ |
| MAML | $\eta = 3 \times 10^{-4}, K = 3$ | $\eta = 5 \times 10^{-1}, K = 3$ | $\eta = 3 \times 10^{-4}, K = 3$ | $\eta = 1 \times 10^{-3}, K = 2$ |
| FOMAML | $\eta = 3 \times 10^{-5}, K = 2$ | $\eta = 1 \times 10^{-2}, K = 5$ | $\eta = 2 \times 10^{-4}, K = 2$ | $\eta = 4 \times 10^{-4}, K = 3$ |
| Reptile | $\eta = 5 \times 10^{-3}, K = 2$ | $\eta = 2 \times 10^{-1}, K = 3$ | $\eta = 4 \times 10^{-2}, K = 2$ | $\eta = 4 \times 10^{-2}, K = 2$ |

## E.3 Ablation study

In this section we introduce the experimental detail of the ablation study. Specifically, the tuning process of the ablation study is as follows: We start from the same setting as the corresponding real experiment in previous section. For example, experiment *Remove graph (FOMAML)* corresponds to HARMLESS (FOMAML). We first use the same learning rate and $K$ as HARMLESS (FOMAML) to perform experiment. If the experiment runs well, we adopt the experiment result. If the training does not converge, we decrease the learning rate and run again.

# References

ACHAB, M., BACRY, E., GAÏFFAS, S., MASTROMATTEO, I. and MUZY, J.-F. (2017). Uncovering causality from multivariate hawkes integrated cumulants. *The Journal of Machine Learning Research*, **18** 6998–7025.

Table 7: Learning rates of experiments of ablation study.

| data type | LR |
|---|---|
| Remove inner heterogeneity ($K = 3$) | 0.1 |
| Remove inner heterogeneity ($K = 5$) | 0.1 |
| Remove grouping (MAML) | 0.1 |
| Remove grouping (FOMAML) | 0.01 |
| Remove grouping (Reptile) | 0.2 |
| Remove graph (MAML) | 0.2 |
| Remove graph (FOMAML) | 0.005 |
| Remove graph (Reptile) | 0.2 |