[Reviews · NeurIPS 2019]

Reviewer 1



The authors propose a mixture of Hawkes process which considers relational information. All event occurrent times of each subject is treated as a mixture of K common Hawkes process, which means the temporal dependency is fully modelled by the Hawkes process. The Hawkes process parameters of subject i have a small deviation to the common K common Hawkes process and the deviation is determined by the likelihood’s gradient (Eq (1)), which can be learned using a method called MAML. The relational information (if subject i and j are linked) is modelled by the weights of the K common Hawkes process for subject i and j. Variational inference (E-step) is used to approximate the posterior of Z and \pi given the parameters of K common Hawkes process and the data. Then the parameters of K common Hawkes process are maximised. The method is compared with other similar approaches and achieves better performance. The idea of incorporating the relational information and subject special parameter (MAML) parts are very interesting. The derivations seem reasonable and correct. I would like to accept this paper if the author could address some questions regarding the experiments. 1. Is it standard to compare the log-likelihood in this field? In my understanding, the author are tackling a prediction problem. Is it possible to predict the expected occurrence time of the next event and compare the RMSE? Also since this method is optimising the log likelihood while others are not, the comparison seems unfair. 2. Why all methods are Hawkes process based? Is it possible to compare with RNN? 3. In line 198-199, it is said that only the last time stamp is taken out. Does all other time stamps used in the training? My understanding is that all time stamps should be taken out for the test data.

Reviewer 2



In this paper, the authors propose a hierarchical Bayesian mixture of Hawkes processes with a parameter adaptation mechanism based on a meta-learning technique for modeling multiple short event sequences with graph-like side information. In the proposed model, each sequence is modeled by a mixture of Hawkes processes, whose mixture ratio has relation to the adjacency of the sequence to the other sequences. Moreover, the parameters of the component Hawkes processes are slightly varied among sequences using the mechanism of the model-agnostic meta-learning framework. The authors provide experimental results on synthetic and real-world datasets, which show the superiority of the proposed method. Overall, the paper is very well written. The technical details are explained in an easy-to-follow way, and the proposed method is clearly positioned in the context of event sequence modeling using Hawkes processes. I am not completely sure about the motivation to use MAML to adapt the parameters to each sequence. What is the biggest advantage of using MAML instead of just fluctuating $\theta_k^{(i)}$ around $\theta_k$ using some regularization terms? Or in other words, isn't it possible to consider the multi-task learning (using graph information) for mixtures of Hawkes processes? The figures in Table 1 would become easier to interpret if the colors of the nodes in the right three columns are roughly aligned to those in the leftmost column geometrically. In Table 2, the performance of HARMLESS with MAML, FOMAML or Reptile greatly differ in some cases. Is there any guideline to choose which meta-learning method should be used in general? ----- [Update after authors' rebuttal] Thank you for the rebuttal. I read it. The points I mentioned are minor (they are just about clarification), so I maintained my score to be 7. Good luck!

Reviewer 3



This paper presents a meta-learning method for learning heterogeneous point process models for short sequence data with a relational network. A hierarchical Bayesian mixture Hawkes process model is proposed to incorporate relational information. The method has been tested on both synthetic and real data. The Bayesian model captures the underlying mixed-community patterns of the relational network. Meanwhile, the model enables knowledge sharing among sequences and facilitates adaptive learning of individual sequences using the model agnostic meta learning technique. A stochastic variational meta-EM algorithm is also derived. My major concern is the performance of the proposed method. In the experiment, the performance of HARMLESS (MAML) is lower than 2 baselines for StackOverflow data. Also, the performance of the 2 HARMLESS methods (FOMAML and Reptile) is much lower than HARMLESS (MAML) for LinkedIn data. The performance of proposed method could be further improved. Moreover, it will be useful to discuss the computation complexity of the proposed method.

[Author Response · NeurIPS 2019]

We thank all reviewers for the constructive comments.

**To reviewer 1:**

**Q1**: It is standard to compare log-likelihood. Examples include Xu et al., (2016, arXiv:1602.04511), Mei et al.
(2017, arXiv:1612.09328) and almost all our cited papers on point process. This is because the prediction of next
arrival is essentially predicting a distribution, and log-likelihood can give a better characterization of the predicted
distribution compared to RMSE. Figure 1 shows a simple example. Generally speaking, RMSE only evaluates the
mean, while likelihood evaluates mean and higher order moments (see consistency theorems in Wald (1949, The Annals
of Mathematical Statistics)).

All baselines in our paper optimize log-likelihood. The comparison is fair. It is natural to first model event sequences as
a probabilistic model, then estimate the model using MLE. Therefore, many existing methods adopt such a procedure.

Figure 1: Simple illustration of why log-likelihood is a stronger metric than RMSE, where (a) shows the ground truth distribution, and (b) shows 3 predictions of (a). In (b), prediction 2 recovers the ground truth perfectly, while prediction 1 and 3 do not. (c) and (d) show the RMSE and log-likelihood estimated from 1000 realizations of the ground truth. RMSE cannot distinguish which prediction is better, while the log-likelihood for the best prediction is the largest.

**Q2**: We chose methods to be Hawkes process-based because we are targeting event sequences that have self-exciting
property. For example, in the MathOverflow dataset, a user that just answered one question usually comments on
several other answers. It is widely accepted that Hawkes process is suitable to model data with such property (Laub et
al., 2015, arXiv:1507.02822, Rizoiu et al., 2017, arXiv:1708.06401).

RNN is not suitable for the short sequence data we are targeting, so we did not include it. There are two possible ways
to incorporate RNN to perform the task:

1. The first one is to adopt an RNN (or LSTM, GRU) to directly fit $\tau^{(n+1)} = f(\tau^{(1)}, \cdots, \tau^{(n)})$. However, one critical
drawback of this model is that it can only provide a point estimation, while the Hawkes process can predict a distribution.
Such distribution is important in our case: (i) the next timestamp is a random variable in nature with a large variance;
(ii) in real applications, the time interval that has larger probability density for next event may be of more interest than a
single estimate. However, naive RNN models cannot give such information.

2. Another option is to adopt neural Hawkes process model in Mei et al. (2017, arXiv:1612.09328). Such a model uses
RNN to parametrize the *evolving* intensity of the Hawkes process, which usually works well for *long* sequences. In our
case, however, the short history cannot provide enough information to fit the complex evolution of the sequences. In our
preliminary experiments, we observed that (i) the training is unstable, and (ii) the performance is not as good as the
standard Hawkes process model (*MLE-Com*).

We will add clarification in the next version.

We want to remark that our HARMLESS framework can actually be complemented by RNN-based models. As we
mentioned in the Discussion section (line 340), if longer sequences are targeted, HARMLESS can naturally extend to
more flexible models by replacing the standard Hawkes process part to Mei et al. (2017, arXiv:1612.09328).

**Q3**: We use all other timestamps in training because HARMLESS builds different models for different subjects, and
thus we need to evaluate each model individually. One should not use the model for another user to predict the future of
this user. In addition, this is the common setting in unsupervised meta learning (Hsu et al., 2018, arXiv:1810.02334).

**To reviewer 2:** The biggest advantage of using MAML is its adaptivity (Grant et al., 2018, arXiv:1801.08930). We will
add more explanation in the next version. We will also adjust the figures and add empirical guidelines for choosing
meta-learning methods.

**To reviewer 3:** We will add more discussion regarding the performance of different models in the next version. In
short, different models are suitable for different data. For example, Reptile is more suitable for larger datasets.

We will add discussions on the computation complexity in the next version. Roughly speaking, if the batch size is $n$,
then the time complexity per iteration is $O(n^2)$.

[Meta-Review · NeurIPS 2019]

The paper proposes a hierarchical model for multivariate point process data with known network information. It uses a mixture of Hawkes processes for the point process observations, and the treats the observed network as a mixed membership stochastic block model sharing the same mixture weights. The main technical novelty is to use model agnostic meta-learning (MAML) to implement the hierarchical prior on the Hawkes process parameters. However, this technical contribution (MAML + Hawkes/network models) is not compared to standard hierarchical Bayesian techniques. Specifically, the parameters \theta_{k}^{(i)} are only three dimensional (background rate \mu, scale \delta, and time constant \omega). The authors should show that MAML is really necessary here; specifically, that textbook hierarchical modeling techniques [like a simple conditional distribution p(\theta^{(i)} | \theta)] do not suffice. R2 briefly commented on this in their review as well. Though the general idea of hierarchical modeling of Hawkes processes and networks is interesting and the submitted paper develops some novel methods for this problem, the omission of standard hierarchical modeling baselines is a critical shortcoming. Despite these serious concerns, the reviewers gave favorable enough scores to warrant acceptance. I would still strongly advise the authors to implement a simple hierarchical modeling baseline before final publication. Other issues: - The latent variable z_i is never actually defined and does not appear in Fig 1, though the reader has to guess that it is an class indicator for \tau_i.